# Five Looks at Emmaus: Revelation, Resonance, and the Sacramental Imagination

Anthony J. Godzieba

Department of Theology & Religious Studies, Villanova University, Villanova, PA 19085, USA;
anthony.godzieba@villanova.edu or agodzieba@verizon.net

**Abstract:** The intersection between religious experience and aesthetic experience has become so obvious that the current "aesthetic turn" in Christian theology no longer needs to be defended. In this essay, I discuss that intersection point from the point of view of Roman Catholicism, in order to demonstrate the bold claim that the arts and the performance they evoke from us are as important as the creed for Catholicism. The essay aims to do three things: first, to examine that intersection point and emphasize the elements of intentionality and desire; second, to analyze one expression of that intersection, namely the connection among Catholic faith claims, the visual arts, and Catholicism's incarnational-sacramental imagination (using depictions of the post-Resurrection Emmaus story); third, to use hints from Hartmut Rosa's recent work on "resonance" to tease out how revelation and transformation occur at this intersection.

**Keywords:** theological aesthetics; aesthetic experience; Catholicism; Emmaus; sacramental imagination; resonance





## 1. Introduction

There is no doubt that there is an intersection between religious experience and aesthetic experience—so obvious, in fact, that the current "aesthetic turn" in Christian theology no longer needs to be vigorously defended. With this in mind, my essay aims to do three things. First, I want to focus on that intersection point and emphasize two common elements, namely intentionality and desire. Second, I want to examine a particular expression of that intersection, namely the intimate connections among Catholic faith claims, the visual arts, and Catholicism's essential incarnational-sacramental imagination. That imagination is a necessary precondition not only for understanding the seven doctrinally defined sacraments in the Roman tradition but also for the believer's reception of the ongoing effects of divine grace beyond the text-based artifacts of the tradition. In other words, the Catholic tradition, like the wider Christian tradition in which it is situated, is what can be termed a *Wirkungsgeschichte,* a "history of effects" (Gadamer 1989, pp. 300–7) an ensemble of confessions, concrete practices, and reflections that mediate and shape the presence of divine revelation. By using examples (particularly works that focus on the post-resurrection Emmaus story in Luke 24:13–35), I want to show how the sacramental imagination puts that intersection point into play. Finally, this essay also looks to Hartmut Rosa's recent work on "resonance" and "uncontrollability" (*Unverfügbarkeit*) for hints as to how one might understand what occurs at these points of revelation and transformation.

By putting these elements together, I want to demonstrate that the arts are as important for Catholicism as the creed, the ancient and central confession of faith recited at every Sunday liturgy. This is admittedly a provocative claim. It is proposed in order to demonstrate that the link between the incarnational-sacramental imagination and aesthetics has much to do with the heart of the gospel message of discipleship—living a Jesus-like life, living one's life according to the values of the Kingdom of God that Jesus Christ preached and practiced.

## 2. What Art Does

Before moving forward, we must take a detour through two crucial discussions of what art actually does.

First, there is the art historian and cultural critic W. J. T. Mitchell's blunt, provocative question "What do pictures want?" (Mitchell 1996, 2005). One answer, he says, is that they want "not to be interpreted, decoded, worshiped, smashed, exposed, or demystified by their beholders, or to enthrall their beholders" (Mitchell 2005, p. 48). In other words, they do not want to be reduced to texts and analyzed like them. Mitchell concedes that interpretive methods provide some help in disclosing the meaning of images. But, he asks, is the "meaning" the most fundamental thing that matters when we encounter a picture? By treating pictures as signs and symbols that need decoding, we end up ignoring the images themselves. Instead, we focus more on those behind-the-scenes intentions that are considered to be the "real" meaning. We thus betray the images themselves. The alternative, Mitchell suggests, is to emphasize the "pictorial turn" rather than the "linguistic turn." In other words, we need to take a more *performative* approach, to allow ourselves to be in the presence of pictures *as pictures* rather than as tokens, to encounter their richness as images, and to be invited and provoked (and seduced, quizzed, entertained, and so on) by them as we experience an *active relationship* with them. We need to focus on the images themselves—from icons to propaganda, from ads to paintings, from photorealism to the most abstract images—and simply ask what they desire from us, which may indeed be different from what pictures "mean".

Despite the power that images are assumed to have in contemporary consumer culture, Mitchell surprisingly argues that pictures "may be a lot weaker than we think" (Mitchell 2005, p. 36). In fact, the "desire" of images is sparked by their need, their emptiness, by what they lack. They need us, the viewers, to complete them. And in this way our encounter with pictures is a lot like our encounter with the sacraments of the Roman Catholic tradition. Sacraments and sacramentality bespeak presence, gift, fullness of grace running over. They desire to give from their plentitude. And surely they "need" us to actualize their potential here and now. They are ritual *performances* that engage us personally, similar to our encounter with images: both invite us into the depth of their "world" and transform us when we accept. The difference, of course, is that the "depth" in sacramentality, namely participation in divine grace, originates in God who makes the first move. The old Catholic sacramental theology of *ex opere operato* (by the work having been done) is a valuable reminder of this, insisting on God's eternal faithfulness to us and inexhaustible gift of divine life to humanity as the motivating force of sacramentality that overcomes any ministerial deficiency. But the fact that the force of pictures comes from their lack, not their full presence, and that they beckon us to fill that lack by participating in the encounter, underlines the *performance* that we enact: it turns looking into seeing into encountering a reality that cannot be reduced to text or code but is most intensely experienced by diving in, by active participation. This gives us the license to call such a relationship "sacramental".

The second discussion is by the literary critic George Steiner. His book *Real Presences* makes a surprising argument: without God, there is no art. To put it more positively, the very possibility of human communication assumes God's presence and, in Steiner's words, "the experience of aesthetic meaning in particular, that of literature, of the arts, of musical form, infers the necessary possibility of this 'real presence'". And so, when we directly experience any form of art whatsoever—"when we encounter the *other* in its condition of freedom," as he says—we make "a wager on transcendence" (Steiner 1989, pp. 3–4). In these encounters with the creative imagination, we catch a glimpse of the divine creativity which makes our own human creativity possible and which artistic creativity mimics. Steiner insists that we must encounter this creative communication directly, in person; every other experience at a distance is a degraded substitute. See the movie, rather than reading the review. Read the novel, rather than listen to someone discuss it. Go to the

concert; don't just read the tweets. Commentary is derivative and parasitic. The presence of transcendence in the artistic wager on God—*that* is what is really real.

Where Steiner clinches his argument is in the book's final two devastating paragraphs where he is not arguing at all, but rather describing our desperate need for art. He uses a surprising metaphor for the intersection of art and life: the Triduum that ends Lent and Holy Week in the Christian liturgical calendar: Good Friday, Holy Saturday, and Easter Sunday. Steiner says that all of us, Christians and non-Christians, believers and atheists, know of Friday: Christians identify its emptiness with the seeming godforsakenness of the Cross, but atheists know of it as well, "of the injustice, of the interminable suffering, of the waste, of the brute enigma of ending, which so largely make up not only the historical dimension of the human condition, but the everyday fabric of our personal lives". Believers and non-believers know of Sunday as well: "To the Christian, that day signifies an intimation, both assured and precarious, both evident and beyond comprehension, of resurrection, of a justice and a love that have conquered death. If we are non-Christians or non-believers, we know of that Sunday in precisely analogous terms. We conceive of it as the day of liberation from inhumanity and servitude.".

> But ours is the long day's journey of the Saturday. Between suffering, aloneness, unutterable waste on the one hand and the dream of liberation, of rebirth on the other. In the face of the torture of a child, of the death of love which is Friday, even the greatest art and poetry are almost helpless. In the Utopia of the Sunday, the aesthetic will, presumably, no longer have logic or necessity. The apprehensions and figurations in the play of metaphysical imagining, in the poem and the music, which tell of pain and of hope, of the flesh which is said to taste of ash and of the spirit which is said to have the savour of fire, are always Sabbatarian. They have risen our of an immensity of waiting which is that of man. Without them, how could we be patient? (Steiner 1989, pp. 231–32)

Art is never merely about the intention of the artist or the enjoyment of the viewer or listener. Art points; it is intentional in the phenomenological sense, always hinting at the "more" that we desire, the hidden and liberating ultimacy that will fulfill us. Our Saturday lives—conglomerations of meaning and absurdity, ecstasy, and tragedy—are never completely devoid of meaning but also never satisfied. We are constantly gripped by what the philosopher Maurice Blondel called the dramatic clash between what we desire and what we in fact accomplish when we act (Blondel [1893] 2003, pp. 314–29). The task of the best and highest art is to give us hope by suggesting in a creative imaginative way our goal, the heights of happiness and fulfilled human potential that can only occur in a deep participation in what is hidden beyond the limits of our lives, the perpetual divine givenness that makes life possible.

And so it is no surprise that the cover of Steiner's book carries a reproduction of one of the "nocturnes" of the seventeenth-century painter Georges de La Tour—scenes of dark, obscure interiors illuminated by shafts of light whose hidden source is the necessary presence that allows the scene to appear at all.[1] In de La Tour's *Christ and Saint Joseph in the Carpenter's Shop* (https://collections.louvre.fr/en/ark:/53355/cl010063817, accessed on 6 July 2023) we have Steiner's aesthetic paradigm: the interplay between the concealed and the revealed. The active presence of the hidden light source is the necessary precondition for the disclosure of the particular figures and their own activity—a performance, if you will, that is the essence of the sacramental.

Mitchell and Steiner both agree that our encounter with art is not passive or static. Despite all appearances, that encounter—with either painting, sculpture, music, or literature—is a *performance*. It includes our active participation in the reality that the work evokes and that we have a role in realizing, very much like the performance of a musical score. That reality of the work is more than any mere surface image, and is most fully revealed only in the midst of that particular performance and the experience of its impact. Mitchell describes what we might call the "horizontal" aspect involving two parties, Steiner the

"vertical" involving three, including the transcendent, which is ultimately the heart of the matter.

### 3. The Creed and the Sacramental Imagination

These performative aspects rely on the activity of the imagination at the heart of the creative process. A few decades ago, the philosopher Richard Kearney, in reviving the key role of the imagination after it seemed to have died under the crushing weight of "pseudo-events" and the commodified consumer image, noted two roles of the imagination: the critical and the poetic (Kearney 1987, pp. 42–43; 1988, pp. 361–97). The critical imagination demystifies the origins of images and carefully discriminates between authentic and inauthentic aspects of the present. Even more importantly, the poetic imagination (in the sense of poiêsis, "inventive") challenges the status quo of endless imitation and objectification by daring to invent new possibilities of existence that break free of any technological and ethical quagmire. Here is Kearney's pivotal definition: "Renouncing the pervasive sense of social paralysis, the poetic imagination would attempt to restore man's faith in history and to nourish the belief that things can be changed. The first and most effective step in this direction is to begin to imagine that the world as it is could be otherwise" (Kearney 1987, p. 44). The imagination is not caught up in pure fantasy, then, but is directly responsive to the reality of our situations and aims for what is different and better than business-as-usual.

What do these commentaries on the action of art and the imagination have to do with Catholicism? When we look at the Nicene Creed (really the Nicene-Constantinopolitan Creed of 381 AD), we must concede immediately its immense dogmatic weight that sets the baseline for what claims count as Christian; after all, it is the product of four centuries of discussion and debate about the reality and significance of Jesus the Christ and the Trinity. But it is obviously an exercise in the poetic imagination as well, a crystallization of four centuries of "thinking otherwise" about the relationships among God, humanity, and created reality.

Nicene-Constantinopolitan Creed (First Council of Constantinople, 381 A.D.)

The Exposition of the 150 Fathers

| | |
|---|---|
| **[A]** | We believe in one God, the Father, the Almighty, maker of heaven and earth, of all that is visible and invisible. |
| **[B1]** | We believe in one Lord, Jesus Christ, the only son of God, eternally begotten of the Father, God from God, Light from Light, true God from true God, begotten, not made, one in Being (consubstantial; homoousios) with the Father. Through him all things were made. For us men and for our salvation he came down from heaven: |
| **[B2]** | By the power of the Holy Spirit he was born of the virgin Mary, and became man. For our sake he was crucified under Pontius Pilate; he suffered, died, and was buried. On the third day he rose again in fulfillment of the Scriptures; he ascended into heaven and is seated at the right hand of the Father. He will come again in glory to judge the living and the dead, and his kingdom will have no end. |
| **[C]** | We believe in the Holy Spirit, the Lord, the giver of life, who proceeds from the Father and the Son. With the Father and the Son he is worshiped and glorified. He has spoken through the Prophets. |
| **[D]** | We believe in one holy catholic and apostolic Church. We acknowledge one baptism for the forgiveness of sins. We look for the resurrection of the dead, and the life of the world to come. Amen. (Tanner 1990, vol. 1, p. *24; with my division into four parts). |

Part A urges us to think otherwise about the relationship of God and reality: reality is not eternal, nor is it a self-starter; the all-powerful God is its creator, but God is no neutral "force" or detached Supreme Being, but can most adequately be talked about in personal, even familial terms ("Father"). In Part B, Jesus the Christ must be seen as proceeding from the Father before all time; in fact, he is everything the Father is except he is not the Father (all the abstract metaphysical language in Part B1, including homoousios, "consubstantial"). But the Son of God is not the aloof Gnostic savior, nor is he simply a divinity in a human costume; he is part of our human history as well (all the historical language of Part B2)

and became human for our sakes. Death did not have the last word for him, nor will it for us. Part C insists that God did not leave the world after the resurrection of Jesus. The Spirit, the personal presence of the Father and the Son, continues to pour out salvific life and offer guidance.

The genius of the Nicene Creed is that it takes up and presents for the believer's assent all three performances of divine presence in reality: Creation and God's presence in the world, done on God's initiative; the Incarnation as the particular intensification of divine presence in Jesus of Nazareth; and the Resurrection, which promises this particular intensification as eternal life for all Jesus' disciples, those who have lived a Jesus-like life and performed the values of the Kingdom of God (which gospels claim is the only way in which those values can be made actual in history). The Creed takes these intense performances of grace and crystallizes them in a rhetoric—partly personalist, partly historical, partly philosophical—that impels believers to approach reality otherwise than some flattened material-empirical construct. Believers approach it as revelation, as God's imaginative otherwise about materiality and particularity: finite creation can mediate our participation in the infinite mystery of God's love and grace, a presence intensified by the Incarnation and guaranteed as the ultimate possibility for each of us by the Resurrection. The incarnational faith of Christianity is the basis for its sacramental imagination.

## 4. Art and Sacramentality

One sees a growing intense awareness of this sacramental imagination in the Catholic religious art of the early modern period, the sixteenth and seventeenth centuries—precisely the period when what counted as "Catholic" and "Christian" was strongly contested. It has been documented by the Jesuit John O'Malley and others that the flowering of Renaissance humanism was accompanied by a strong emphasis on the Incarnation in all aspects of Catholic life: devotional, liturgical, intellectual, and aesthetic (O'Malley 1974, 1996, 2000). (In fact, one can trace vestiges of this emphasis all the way into the Catholicism of the 1950s.) This contrasted strongly with the more pessimistic assumptions about human nature in the late medieval period. Instead of an anthropology that emphasized a fallible humanity predisposed to evil, the humanists favored the view that saw human nature as perfectible and open to strategies of persuasion. This frame of reference reflected the clear theological choice not to take the Fall of Adam and Eve narrated in Genesis as one's anthropological starting point, but rather to choose God's act of creation, in which the person "was dignified by the unique status of the imago Dei" (Bradshaw [1991] 2004, p. 115). As O'Malley puts it, "what better way could there possibly be to drive home the truth of the dignity of human nature than to insist that God himself had not disdained to assume it and had, indeed, become man—truly man" (O'Malley 1996, p. 214).

The works discussed in this section were chosen (out of many possible choices) because they articulate clearly a fundamental claim of the incarnational-sacramental imagination, namely that the finite mediates the infinite, that nature mediates grace. In other words, these examples of early modern religious art portray—and indeed perform—the chiasmus of the natural and the supernatural, the visible and the invisible with an intense focus on creation, especially human embodiment, as the means of that portrayal.

Some of the roots of this incarnational emphasis lay in late medieval affective piety and the images that supported it. For example, Rogier van der Weyden's altarpiece *Deposition of Christ from the Cross* (c. 1430–35; https://www.museodelprado.es/en/the-collection/art-work/the-descent-from-the-cross/856d822a-dd22-4425-bebd-920a1d416aa7, accessed on 6 July 2023) crams the figures together as a kind of living tableau in what seems to be a golden case (de Vos 2002, p. 76). The cross is reduced in size; it is the after-effects of the Crucifixion that seize our attention, especially the grief-stricken faces of the mourners. The swooning Mary seems almost crushed by the death of her son. It is no accident that the position of her body parallels his; she identifies with his suffering. Except for Joseph of Arimithea who tenderly holds Jesus' lifeless body, the head of every figure is tilted, echoing that of Jesus, and grief ripples outward toward the painting's extremities. That



grief is inconsolable and deeply affecting. The weeping woman at the far left, seemingly outside the enclosure marked by the "parentheses" of the bent bodies of John (left) and Mary Magdalen (right), represents us, the onlookers. Though not part of the original Crucifixion event, we are nonetheless invited to participate and feel along (compassio) with the privileged eyewitnesses—we become eyewitnesses as well. Rogier's tableau perfectly reflects "the ideal of imitating Christ in all respects" and the "passionate devotion to [Christ's] humanity" that marked the medieval period from the eleventh century onwards.[2]

Early modern Catholic art, especially in the Baroque period, took up both the incarnational imagination and this embodied spirituality and coupled it with the Baroque theatricality that breaks down the barrier between the spectator and the work, all in service to seeing reality as a theatrum sacrum. It is true that the Baroque fascination with artifice and ruin reflects an underlying sense of distance between this visible "vale of tears" and the realm of the Beatific Vision.[3] But music, art, and architecture represented a sacramental mediation, a bridge between the visible and the invisible. The visible was consistently portrayed as vacillating between reality and appearance, truth and illusion. This did not deny the perduring presence of God's goodness in creation nor the perduring ontological effect of the Incarnation in sanctifying history and dignifying the human person. Rather, it was a common-sense acknowledgment that this grace-filled world is also the temporal, finite, unfulfilling and unfulfilled stage-set of human life, whose possibilities can be realized only by its supernatural destiny, a destiny that can be suggested or hinted at, perhaps even tasted to a small degree, but never savored in its fullness until after the final judgment.

Gianlorenzo Bernini supplies classic examples of this sacramental view. *His Ecstasy of St. Teresa* (1647–1652; https://artincontext.org/wp-content/uploads/2022/01/Ecstasy-of-Saint-Teresa-by-Gian-Lorenzo-Bernini.jpg, accessed on 6 July 2023) is famous for a reason: Teresa of Avila's mystical experience of "transverberation" in prayer is played in an erotic key that matches perfectly Teresa's own ecstatic report and masterfully solves the aesthetic problem of how to externalize internal states of consciousness (Teresa of Avíla [1562–1565] 1987, p. 252). Bernini's choice of erotic embodiment encompasses both states (https://www.wga.hu/cgi-bin/highlight.cgi?file=html/b/bernini/gianlore/sculptur/1640/therese1.html&find=Teresa, accessed on 6 July 2023). Even more intimate and intensely erotic is one of his last works, *Blessed Ludovica Albertoni* (1671–1674; https://upload.wikimedia.org/wikipedia/commons/b/bc/Bernini_Ludovica_Albertoni_2.jpg, accessed on 6 July 2023). The eroticism of that intimacy—more active, more convoluted, and more private than that of Teresa—shocks us with its agitation. But again, the effect of this affective charge is sacramental: the physical consummation suggested here is symbolic of the mystical consummation of divine love that Ludovica experienced in the rapture before her death, a consummation that "is shared sacramentally by all who partake of the Eucharist at her altar/tomb" (Scribner 1991, p. 118).

Similar intensities and effects without the intense eroticism are conjured up on frescoed church ceilings. The Jesuit Andrea Pozzo's wonderfully exuberant fresco for Sant' Ignazio in Rome, *The Apotheosis of St. Ignatius* (1691–1694; https://jesuits-eum.org/wp-content/uploads/sites/60/2018/06/volta-sant-ignazio-roma.jpg, accessed on 6 July 2023), with its architectural illusions, flying figures, and the heavenward ascent of St. Ignatius, the founder of the Jesuits, fools the eye into thinking that the flat ceiling truly opens out to heaven three-dimensionally.

Pozzo's frescoes were famous throughout Europe; his treatise on perspective was widely used and his methods were copied in numerous Catholic churches throughout southern Germany and Italy into the late eighteenth century. Johann Jacob and Franz Anton Zeiller's frescoed dome at the crossing in the Benedictine abbey church in Ottobeuren is an example of this (1757–1758; https://www.sueddeutscher-barock.ch/Bilder_jpg/werkbild/o/Ottobeuren_Ki-6Gr.jpg, accessed on 6 July 2023). It offers a glimpse of heaven during the descent of the Spirit on Mary and the apostles at Pentecost; the piercing of the limit is performed by a downward motion of God who invites us to ascend upward. An even more spectacular example is the tableau of Mary's assumption by Egidius

Asam (1723), behind the main altar in the Benedictine abbey church in Rohr, north of Munich, demonstrating for believers that the distinction between heaven and earth is thin at best (https://commons.wikimedia.org/wiki/Category:Kloster_Rohr_(Bayern)#/media/ File:Kloster_Rohr_in_Nby_Detail_Hochaltar.JPG, accessed on 6 July 2023). Both churches make the point that when in this particular place, in communion with the church and receiving the body and blood of Christ (the Ottobeuren dome is directly over the base of the altar steps, where the Eucharist is given), one has access to the grace needed for salvation.

Each of these works is a performance in the authentic sense of that word, and in the ways discussed by Mitchell and Steiner. If we ask Mitchell's question "what do pictures want?", in these cases they want us to complete them by accepting their appeal to resonate emotionally with them, be swept up by them either subtly (Rogier's Deposition) or dramatically (Bernini, Asam's Assumption). If we take Steiner's point, each work—again either subtly or dramatically—makes us ask "why am I so affected? why so swept up? what desire in me is this beauty evoking?" And the response would be that these ways of imagining otherwise provoke us to imagine otherwise and indeed to imagine our selves otherwise, to seek an even richer experience beyond what counts as normal, and to wonder about the reason for our perpetual desiring and what its ultimate target is. In more Catholic terms, each performance of the incarnational and sacramental imagination draws the observer into that subtle dynamism where the finite mediates the infinite, nature mediates grace. We do not have access to holiness despite our embodiment, but because of it— because of the graced possibilities bestowed on it at creation, intensified at the Incarnation, guaranteed by the Resurrection of Christ, and made actual only by our performance of discipleship.

## 5. Five Looks at Emmaus

I want to focus on the intersection of the incarnational imagination, art, and discipleship by taking a specific case, the evangelist Luke's moving narration of one of the pivotal post-Resurrection appearances of the risen Jesus, his encounter with Cleopas and his unnamed companion as they journey disconsolately toward the village of Emmaus (Luke 24:13–35). We know the story, and we anxiously await its climax when we hear it read at the liturgy as the gospel of the day. Being "on the way to Emmaus" has served many theologians and spiritual writers as a metaphor for discipleship and the spiritual life—the journey from despair to faith and hope through a salvific encounter with Christ, especially in the Eucharist. The Emmaus narrative continues to be an immensely rich resource for reflection upon the practice of "following Jesus." What follows are five "looks" or "takes" where artists have depicted key points of the narrative, specifically the disciples' encounter with Jesus, his interpretive conversation with them, and the unexpected epiphany at the evening meal.[4]

Jesus' encounter with Cleopas and his unnamed colleague has been depicted in some very traditional ways. Take as an example the *Landscape with Christ and his Disciples on the Road to Emmaus* (1640) by Jan Wildens (landscape) and Hans Jordaens III (figures) (https://upload. wikimedia.org/wikipedia/commons/1/15/Jan_Wildens_-_Landscape_with_Christ_and_his_ Disciples_on_the_Road_to_Emmaus_-_WGA25745.jpg, accessed on 6 July 2023). The figures are posed classically, in a style reminiscent of Renaissance and Baroque depictions of peripatetic Greek and Roman philosophers with their disciples. Since Wildens was known for his landscape paintings, it is no surprise that there is more interest in the lush landscape than the almost accidental figures. The majestic trees and the gorgeous rolling hills in the distance are the first focus of our attention, followed by the discovery of the fisherman on the left, the cottage tucked among the trees, and what looks like a villa peeking through the foliage. A performance, yes, but at a low boil. But even this very classic scene displays a hint of the sacramental imagination: an "ordinary" landscape is revealed to be truly extraordinary, beauty after beauty, small and large, piled up for our view. Since the viewer knows the Lukan story and can discern the painting's clues, it is a perfect setting for the ordinary everyday conversation among strangers that will become an extraordinary disclosure of grace.

In my view, that conversation in Luke's gospel is more angst-ridden than portrayed here. And that is why I want to step out of our chronological order at this point and highlight Georges Rouault's *Les disciples* (1936; https://risdmuseum.org/art-design/collection/road-emmaus-20079813, accessed on 6 July 2023). Here, all detail is stripped away: the fields and lakes on either side of the road are blotches of green and blue, and the village of Emmaus is one formalized red house in the far distance at the end of a mottled red road under an ambivalent sky. The cruciform pattern is unmistakable. The figures are in the midst of what looks like an animated debate. Cleopas and his unnamed companion have been trudging from Jerusalem to what is probably their home village (cf. Fitzmyer 1985, p. 1559), and are not the only members of the Jesus-movement to give up and return to their earlier lives. Luke shows them deep in conversation, even contentious debate, weighed down with disappointment and gloom (*skuthrōpoi* [24:17]). They meet a stranger who joins their journey and their animated discussion. Clearly frustrated, Cleopas begins to pour out their anxieties. Why did the events in Jerusalem leave them in such a despairing state? The answer is found in the intersection of religion and politics that seemingly brought the Jesus movement to a brutal, horrifying end. Jesus' followers have been shocked not only by its violence but by its abrupt shattering of all their expectations. Add to this bewildering stories being circulated that Jesus has been seen alive. With an economy of means, Rouault forces us to focus on the debating travelers; nothing else should attract our attention. Perhaps this is the moment just before Jesus' rebuke of their gloomy misinterpretation of events: "'Oh, how foolish you are!'," he says, "'how slow of heart to believe all that the prophets spoke! Was it not necessary that the Messiah should suffer these things and enter into his glory?' Then beginning with Moses and all the prophets, he interpreted to them what referred to him in all the scriptures" (Luke 24:25–27). This itself is a small epiphany: the Scriptures are not self-evidently clear; they need a contextual interpretation, especially when the context is Jesus' preaching of the Kingdom of God. Using Mitchell's terms, Rouault's image cries out for completion, for an interpretation that fills in how the disciples get from the disaster in Jerusalem to the grace in Emmaus by means of their encounter with Jesus.

The key to the narrative, of course, is the meal shared by the travelers after the disciples urge the still unrecognized Jesus to stay with them "for it is nearly evening and the day is almost over" (Luke 24:29). "And it happened that, while he was with them at table, he took bread, said the blessing, broke it, and gave it to them. With that their eyes were opened and they recognized him, but he vanished from their sight" (Luke 24:30–31). This is the breathtaking moment in Luke's narrative where the transfiguration of the disciples reaches its climax. Their epiphany—that this truly is the risen Jesus—happens in the midst of one of the most intimate moments of personal relationship in ancient Near Eastern culture, the sharing of a meal. Without the invitation to remain in the disciples' company, without the blessing and sharing of bread, without this bond among the three travelers, the veil would not have lifted, the disciples' expectations would not have been transfigured, Jesus would not have been transfigured in their eyes and recognized as loving savior and Lord, and communion would not have been established.

Two earlier masters, Rembrandt and Caravaggio, have approached this pivotal scene with different emphases. Rembrandt painted it twice, the first early in his career in 1629 (https://www.musee-jacquemart-andre.com/en/works/supper-emmaus, accessed on 6 July 2023). He signals the revelation of the identity of the risen Jesus by showing an awkwardly posed Jesus in silhouette, brightly back-lit, with one disciple kneeling at his feet (barely visible, with a toppled chair on the floor), the other in shock at the recognition. The emphasis is clearly on the revelation of the person of the risen Christ and his divinity; the meal is an afterthought.

Rembrandt painted the scene again in 1648, but much differently (https://collections.louvre.fr/en/ark:/53355/cl010062023, accessed on 6 July 2023). Jesus sits facing us in an alcove under an arch, softly glowing, while the disciples' reactions are muted.[5] Again, the meal is barely significant (in fact they seem to have finished); the focus again is on the revealed identity of the risen Jesus, but this time emphasizing his humanity. In both cases,

Rembrandt reduces the complexity of Luke's narrative to a single point, the revelation of the risen Jesus. And thereby the action of the sacramental imagination is focused on one aspect as well: in the 1629 version, the natural mediates the supernatural by being rendered as odd, awkward, almost explosive. In the 1648 version, the natural mediates the supernatural by being more natural, an intensity that seems to well up from within Jesus himself, suffusing his surroundings with a soft light that could symbolize the subtle outpouring of grace upon all.

How differently does Caravaggio react to the scene earlier in the century! In *The Supper at Emmaus* (1600–1601; https://www.nationalgallery.org.uk/paintings/michelangelo-merisi-da-caravaggio-the-supper-at-emmaus, accessed on 6 July 2023) he has chosen to portray the extraordinary within the ordinary, the supernatural within the everyday, by means of heightened concentration upon the ordinariness of the scene and its elements, with almost the precise observational skills of a natural scientist. The work is as luminous as it is virtuosic. Christ is seated at the center, as is usual in Italian tradition. But there is nothing usual about the rest of the composition: its richness almost defies description, and its visual immediacy involves the viewer directly in the experience. The drama is achieved through the contrast of deep dark colors and brilliant highlights, by the sensuous fullness of the figures, and by a dazzling use of perspective. The table is set close to the viewer, but with enough room so that the disciple on the left can be seated before it. That disciple grasps the arms of his chair as if he is about to leap out of it in utter astonishment at the dramatic revelation of Christ and push it through the picture plane. The basket of fruit perched precariously on the edge of the table looks as if it will drop into our laps at any moment. The disciple on the right, wearing the pilgrim's shell, throws out his arms in a gesture of amazement (perhaps symbolizing the crucifixion). His left arm, extended toward the viewer and also seeming to break the picture plane, unites the disciples with us, who also stand amazed.

In this ensemble, all attention is focused on Christ, whose beardless face is perhaps Caravaggio's solution to the problem of why the disciples "were prevented from recognizing him" (Luke 24:16) on the road. Christ's face is illuminated by light falling from the left; its radiance is highlighted even further by the shadow cast by the uncomprehending innkeeper on the wall behind, almost as a kind of dark halo. The light is of divine origin and signals God's transfiguration of this ordinary scene and the revelation of the identity of Christ at the moment the meal is blessed and shared. And how that scene is transformed: a delicate light passes through the water pitcher and casts both water-reflected light and shadow on the table cover in a scene of incredibly eloquent brilliance. The same light is reflected from the table cover onto the face of the disciple on the left, allowing us to see and know his reaction even though we can catch only the barest glimpse of his face.

Rembrandt and Caravaggio are paired here not merely because these works are classics of early modern Western religiosity, but because their differing emphases also illustrate differing responses to the incarnational-sacramental imagination. While I emphasize this imagination as a structural principle of Catholicism, the wider Christian belief in the doctrine of the Incarnation means that at least some aspect of that imagination will animate a religious tradition that confesses Christ's divinity and humanity. Their confessional leanings—Rembrandt toward Calvinism, Caravaggio toward Counter-Reformation Catholicism—shape their responses. "While Catholic religious art presented the Christian mystery in an emotionally evocative fashion, Protestantism favoured storytelling and explanations over a direct appeal to sentiment" (Westerman 2000, p. 274). In either of Rembrandt's portrayals, the focus is Christological, and the sacramental aspect of the meal is downplayed. "By exploring the personal nature of the viewer's relationship to Christ, Rambrandt created new possibilities for devotional art in a predominantly Protestant culture" (Westerman 2000, pp. 274–75), in keeping with the reformers' notions of salvation. On the other hand, the fundamental interpretation of the Emmaus episode by Counter-Reformation Catholicism is eucharistic. Caravaggio's portrayal "preaches" the Eucharistic meaning of the episode and manifests the sacramental imagination and care for ordinary

materiality that he shared with the popular Catholic spiritual reform movements of his time. Indeed, to borrow the title of Arthur Danto's classic work on the philosophy of art, we witness "the transfiguration of the commonplace" (Danto 1981). Caravaggio's complex composition choreographs our performances as viewers. As our eyes move around the picture, the ordinary, without giving up one bit of its materiality, is slowly revealed to be extraordinary; by means of the natural, the supernatural is revealed—to us as well as to the two disciples. The disciples finally recognize Jesus in the blessing and breaking of the bread—thus in a shared relationship that becomes, by the sheer incarnational initiative of God, the precondition for the revelation of Christ's identity and the meaning of salvation.

Finally, to plant us firmly again in the everyday, there is Pieter Aertsen's *Kitchen Scene with Supper at Emmaus* (1603; https://collections.mfa.org/objects/100690, accessed on 6 July 2023). Aertsen was known for his "inverted" scenes of still life which he took over from the Italian Mannerists, a paradoxical style where he dignifies "low" or common life by depicting it in a classical "high" style (Snyder 1985, p. 445; Falkenburg 1995). So here, in the midst of a bustling kitchen full of everyday activities and spectacular fish, the revelation of the risen Christ to the disciples occurs in its own distinct space far in the background, yet in the midst of the ordinary life that goes on around it. We might call this a "Caraviggist" sacramental imagination in a different key: an intense concentration on the ordinary reveals it to be extraordinary, and the supernatural does occur within the natural—but so subtly that you might miss it unless you were looking for it. Though less virtuosic than Caravaggio, Aertsen gives a sophisticated portrayal of the sacramental imagination nonetheless by emphasizing the divine "discretion" that is also part of the fundamental structure of creation: God pulls back to let creation be (Duquoc 1992, p. 3). While still theonomous (grounded in God), the non-divine is granted autonomy as well. We are coaxed by the kitchen scene to discern the divine presence hidden amongst what look like merely everyday concerns.

These Emmaus examples summarize key elements of the Catholic sacramental imagination. It is the way of envisioning reality through the eyes of faith that recognizes that all aspects of created being can mediate grace—the finite mediating the infinite. The framework of the "otherwise" can provide a way of bringing this Catholic commitment to the structure and process of sacramentality into conversation with contemporary culture and also with those who are suspicious of religious metanarratives. With this imagination, believers probe the immediate situation for new possibilities of existence in the light of God's relationship with humankind and with the cosmos. Animated by the revelation of God in Jesus Christ, the Catholic sacramental imagination holds that life can be otherwise than an unending cycle of desires and the commodified images that fulfill them and that God wills that it be so: a life-enhancing otherwise that begins in the present and is fulfilled in the future. This active reconfiguration of the world toward eschatological fulfillment becomes meaningful within the believer's act of faith in God—that is, by imagining as possible the grace and power of God within reality, by the belief in the suitability of the world to mediate such transformation, and by the performative action that brings believer, world, and divine grace together into effective realization.

## 6. Resonance and Revelation

This performative encounter with the aesthetic artifacts of a religious tradition is not merely "religious" or "spiritual," even if it eventually points in that direction. As both Mitchell and Steiner emphasize, it is an intensification of the everyday experience of perception that results in an active, transformative relationship. One result is to recognize that this is due not only to the agency of the embodied "I" but also the agency of the work that "gives" its presence in the encounter.

Maurice Merleau-Ponty's existential-phenomenological approach to perception and embodied subjectivity provides a way of articulating what happens in this encounter. His "phenomenology of perception" builds on both Edmund Husserl's theory of consciousness-as-dynamic-outreach (i.e., intentionality and constitution) and Martin Heidegger's early re-

orienting of Husserl's transcendental idealism toward "everydayness" with his "existential analytic" (Merleau-Ponty 1962; Godzieba 2021). Our consciousness awareness, he argues, is intentional: it streams outward from the embodied self toward what is beyond the self, in order to reveal and know this "non-I" through our perceptual relation to it. Most importantly, the body knows the world before reflection clarifies that knowledge, and in doing so reveals the truth of the world. The key is the revelatory nature of this active relation. There is no radical separation of embodied human subjectivity and the world. Rather, he says, I discover myself as "a subject destined to the world" (Merleau-Ponty 1962, p. xi), an effective involvement of subject and world together in a meaningful whole (*Gestalt*). The "normal functioning" of perception "must be understood as a process of integration in which the text of the external world is not so much copied, as composed" (Merleau-Ponty 1962, p. 9.)[6] My consciousness is radically situated and tied to bodily experience; it never drifts off to a pure realm of abstract universals through which I might know others and the world. Consciousness only gets "filled out," only grasps meaning by its very openness to "what is not myself." The retreat to some detached knowledge of essences in order to know the world and others is a false step.

> All knowledge of man by man, far from being pure contemplation, is the taking up by each, as best he can, of the acts of others, reactivating from ambiguous signs an experience which is not his own, appropriating a structure . . . of which he forms no distinct concept but which he puts together as an experienced pianist deciphers an unknown piece of music: without himself grasping the motives of each gesture or each operation, without being able to bring to the surface of consciousness all the sediment of knowledge which he is using at that moment. Here we no longer have the positing of an object, but rather we have communication with a way of being. (Merleau-Ponty 1964, p. 93)

All my knowledge comes from lived experience which, like the playing of the pianist, ferrets out and discloses the truth of the world while I am involved in experiencing it, and never apart from that involvement. That involvement does not close me off, but rather opens me to the other, to things, and to the totality of meanings we call "world." Meanings are constituted by the interaction of consciousness with the "stuff" of the world, primarily through the "lived body" (*le corps vecu*), and then through reflection (the *cogito*). My immersion in the world through perception already automatically ensures that from the start my existence is double-edged: it is absolutely individual because of my own perspective, rooted in the perceived world by my body; at the same time, it is absolutely universal, since the interplay between my subjectivity and the world opens me to the whole world. It is this accessibility through my embodied standpoint to what is not myself that serves to ground my appropriation of meaning or structure. Even though consciousness accomplishes the constitution of meaning, it can do so only along the lines of the direction or hint provided by the world that is "found ahead of us, in the thing where our perception places us, in the dialogue into which our experience of other people throws us by means of a movement not all of whose sources are known to us" (Merleau-Ponty 1964, p. 93).

The sociologist Hartmut Rosa has taken these insights (among others) and extended them with his recent work on "resonance" (Rosa 2019). It is first of all a sort of response to the "social acceleration" and the "presentist" erasure of time and duration that earlier he had diagnosed as characteristic of our contemporary context (Rosa 2003, 2013; Godzieba 2018, pp. 287–89). More specifically, his theory of resonance offers an antidote to our default late modern view of the world as a "point of aggression," our desire to make the world, seen merely as a series of exploitable objects, "controllable at every level—individual, cultural, institutional, and structural" (Rosa 2020, p. 4). This control always eludes us. The world becomes "distorted" and alien—it "withdraws from us, becoming mute and unreadable. Even more, it proves to be threatened and threatening in equal measure, and thus ultimately constitutively uncontrollable" (Rosa 2020, p. 19). This alienation blocks us from experiencing the richness of life and provokes fear, despair, and the "impotent political aggression" that marks the contemporary world (Rosa 2020, p. 4). "Yet it is only in

encountering the uncontrollable that we really experience the world. . . . Our lives unfold as the interplay between what we control and that which remains outside our control, yet 'concerns us' in some way. Life happens, as it were, on the borderline" (Rosa 2020, p. 2).

"Resonance" is a way of articulating a non-alienating, life-giving relationship between the embodied subject and the world. Resonance presupposes our embodied situatedness and our immediate perceptual response to the world (Rosa 2019, p. 258). Resonance is "a specific mode of relation—i.e., a specific form of being-related-to-the-world—in which the world or at least some segment of it is experienced as responsive . . . " (Rosa 2019, p. 169). That relationship "is formed through af→fect and e→motion, intrinsic interest, and perceived self-efficacy, in which subject and world are mutually affected and transformed." In this relationship, "both sides speak with their own voice . . . Resonance implies an aspect of constitutive inaccessibility" and thus uncontrollabilty (Rosa 2019, p. 174).[7] This relationship has four essential characteristics (Rosa 2020, pp. 32–38):

(1)  "Being affected": being "'inwardly' reached, touched, or moved" by the appeal issued by, e.g., "another person . . . a landscape, a melody, or an idea";

(2)  "Self-efficacy": our capability of responding to the call, "reaching out toward that which moves us" and thereby affecting it (thus, the intensification of a reciprocal relationship);

(3)  "Adaptive transformation": when we are transformed to any degree by the encounter, a change in the way we relate to an other or to the world;

(4)  "Uncontrollability": there is no method for resonance, and no closure; "whenever it occurs, we are transformed; but it is impossible for us to preduct how exactly we will be changed and what the end result of the transformation will be".

Rosa's sketch of the essential elements of resonance reads almost like a map of the affects and effects described by Mitchell in our encounter with images, by Steiner regarding the eschatological hope that the "real presence" of art engenders in us, and by Catholic sacramental imagination regarding not only its claim that the finite mediates the infinite, but that the grace given on God's initiative relies for its full actualization on its active reception by the believer-as-disciple. On one level, the performative response that results from an encounter with any sort of art activates a relationship, much like the performative response of the believer's encounter with the sacraments in the Catholic tradition. And, if we follow Steiner's argument (if there is no God, there is no art), there is also a deeper, more fundamental reality to be recognized: our encounter with art of any type is already "sacramental" because that relationship exposes us to an "other" creative impulse that approaches us and opens out to us, an aesthetic impulse that imitates and presupposes, in turn, the origin of all creative impulses—it is a hint or clue of divine transcendence. Our recognition of this transcending presence can be construed as a kind of "natural theology" that responds to the revelation of grace and gift that we experience as the basic structure of reality. This is not natural theology in the textbook sense of "rational knowledge of God without special revelation," but rather the deeply Catholic sense of a reflection on experience that seeks to identify "the natural 'access-point' of faith" (Kasper 1980, p. 20). In other words, human experience by its very nature participates in a dynamic movement toward God which can be even more fully articulated through faith in God's gift of further self-revelation in and through created being—the claim that is at the heart of the sacramental imagination. If we are able to demonstrate how our finite everydayness is open to transcendence by exceeding its limits, as a condition of its own intentional structure, then we have a chance to render plausible, as far as it is possible, transcendence from "our side", sketching out that "access-point" where the revelation of God meets the embodied self's transcending yearnings and intentional strivings for fulfillment. The intentional thrust outward into the world (a role enacted by the body) and into the future (by the imagination) function as analogs for faith as intentionality, our uncontainable seeking for fulfillment in God. Our faith intentionality finds its counterpart in the intentional thrust "inward" and towards the future on God's part, in revelation, at points of varying intensities. The

peak revelational intensities of creation, Incarnation, and Resurrection give the believer a glimpse at the divine poetic imagination, God's "otherwise".

## 7. Conclusions

Both the Creed and the visual arts can be taken, then, as crystallizations of this imaginative "otherwise" meant to move us to aspire to thêosis ("divinization"), participation in the divine presence that underlies all reality. Both remind us that the Incarnation opens up the materiality of the particular as the arena of this receptivity. The Creed does so as a set of propositions: partly abstract and ontological, partly historical, but wholly confessional, meant to move the confessor more deeply into discipleship. The visual arts as such (if we agree with George Steiner) and Catholic religious art, in particular, do so as invitations to performance: they invite—and indeed provoke—us to experience their performance of depth and meaning beyond the limits of the surface by participating in that performance. These are indeed imaginative revelations that echo the original divine creative impulse and aspire to participate in it. For without this sacramental insight opened up by the arts, this divine clue of grace and fulfillment in a world that is a mixture of pleasures and pains, ecstasies and tragedies—and indeed, a world at times seemingly awash in hatred and violence without end—how could we be patient?

**Funding:** This research received no external funding.

**Data Availability Statement:** No applicable.

**Conflicts of Interest:** The author declares no conflict of interest.

## Notes

[1]    On de La Tour's nocturnes, see (Conisbee 1996).

[2]    See (Constable 1995, p. 179): "The ideal of imitating Christ in all respects deepened in the eleventh century into a passionate devotion to His humanity, which increasingly excluded other models and established Christ as the supreme exemplar for devout Christians."

[3]    See (Harries 1983, p. 179): "The gap that has opened up between the visible and the spiritual, between picture and reality, makes it impossible for the church of the counter Reformation to simply return to the medieval understanding of the church as a more or less literal representation of the divinely established order of the Heavenly City or the cosmos."

[4]    My approach is inspired by the title of Olivier Messiaen's suite of piano pieces *Vingt regards sur l'Enfant Jésus* ("Twenty Gazes upon the Infant Jesus" "Twenty Glances at . . . "): twenty different musical meditations on the Christ Child, written in Paris in 1944 while the city was on the brink of collapse under Nazi occupation.

[5]    Having seen this painting in person during the 2011 exhibition "Rembrandt and the Face of Jesus" at the Philadelphia Museum of Art, I saw how in a roomful of masterpieces it commanded everyone's immediate attention.

[6]    This is Merleau-Ponty's more existential version of Husserl's theory of constitution.

[7]    The neologisms "af→fect" and "e→motion" are Rosa's, signifying the dynamic interrelationship that mitigates any rigid distinction between the embodied subject and the world (although the "constitutive inaccessibility" of each maintains some distinction of subject and world).

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
