# Peer review of "Five Looks at Emmaus: Revelation, Resonance, and the Sacramental Imagination"

_religions, doi:10.3390/rel14070895_

Round 1
Reviewer 1 Report
It is a pleasure to read this article. While it does not bring any major new discoveries to the theory of the image or the phenomenology of art, it interprets their basic ideas accurately and, above all, applies them in a beneficial way to the field of Catholic Eucharistic theology. I particularly appreciate the emphasis on the performativity of images within particular sacred sites or rituals, as well as the emphasis on the corporeality of religious experience as mediated by the perception of the image. The interpretations of specific paintings are presented very convincingly. In my opinion, the study can be published without revisions.
Just minor typos. The paper reads well.
Author Response
Thank you for your postive comments. The paper, in truth, is tracing the incarnational-sacramental which is a wider topic than eucharistic theology. But your points are well taken and I appreciate them.
Reviewer 2 Report
Given the by now clear connection between religious experience and aesthetic experience, the author considers (specifically in a Roman Catholic context) how performance, and the arts are just as important as the time-tested creeds of the faith. This provocative claim is intended to be supported by three movements: considering the intersections of how intentionality and desire are expressed in artistic presentations, exemplifying that intersection in considering the sacraments and the visual arts in Catholicism, and displaying the theoretical application of Rosa´s concept of resonance to consider revelation and change, in the context of these artistic and performative contexts.
There is much to be gleaned from this paper - - non-convoluted theories of art, image analysis, well-written prose, and a short demonstration of how Rosa´s work might connect to the arts. The paper is incredibly well written, researched, and thought-out. The paper does not take an either|or approach to arguing for|against concepts vs images, and it demonstrates the continued importance of creeds. Art does not take their place. With these clearly positive aspects in mind, there are a few concerns the paper needs to address or clarify.
1. Why only Catholicism? Given the author´s own rather ecumenical depiction of Christianity in the opening pages (“…the heart of the gospel message of discipleship—living a Jesus-like life, living one’s life according to the values of the Kingdom of God that Jesus Christ preached and practiced”), it seems the claims are more broadly applicable to most Christian movements. The continued claim made that God “makes the first move” in terms of how images and faith-filled grace are to be received, and how we develop our performative response, jives quite well with most Christian movements.
As the author specifically mentions in the final concluding paragraph “Catholic religious art” this makes the paper´s catholic focus even more difficult, especially since the author uses Rembrandt so much. To my understanding, Rembrandt was Protestant, and if the author is trying to make a specifically Catholic claim about Catholic art, this does not quite work.
One particular use of Merleau-Ponty on embodiment in the paper seems at points contradictory to the reliance upon tradition and the “passing on” of images in the Catholic Church. As mentioned by the author in the paper “All my knowledge comes from lived experience which, like the playing of the pianist, ferrets out and discloses the truth of the world while I am involved in experiencing it, and never apart from that involvement.” (P 22, line 56). That is quite fine, and Christian theology holds personal and individual experience definitely essential to theological development. But by no means is it the only source, and phenomenology more broadly would be willing to accept that. We have reason, tradition/history, and the Bible, etc etc, all of which do not always coalesce with what we experience. Again, a full emphasis upon experience only seems to go against especially the importance of tradition we find in the Catholic church.
Further, I understand the importance of specificity, and I also understand that the Catholic church has a particular bend towards images and aesthetic experience that perhaps, for example, one would not find in a Baptist Church service in an underground church in China. However, most of the context and examples used in the paper could just as easily be applicable to other Christian movements. The many references to “Catholic” in the paper at times serves to appear more exclusivistic, and that seems to go against what appears to be the intentions of the author in the paper - - to demonstrate a broader, performance based religiosity. It of course would be fine for the author to choose to remain committed in their analysis to Catholic faith. However, all of these points need to be taken into consideration, if only to avoid the exclusivistic tone, the problematic aspects with Rembrandt and the MP quotation, and to make the paper a bit more accessible to readers who are from different denominations.
2. It is unclear what the author means by aesthetic experience and “art”. Are these claims applicable to all forms of art, or are they limited to those used in this paper? Could it be that Christianity (namely, the incarnation) helps get beyond the critical imagination, and support more so the poetic and “sacramental” imagination via creation?
3. Similarly, are all religious images actually “art”? Surely any image can be a conduit by which one might have an experience with God. But many instances of reproduced images or childrens drawings may not bear the same weight as Rembrandt. Further, I can certainly imagine the necessity of the aesthetic experience to Christian life and faith. What becomes more difficult to understand is the necessity of imagery for this, images that most often than not are in repeated forms and icons that are not “original” artistic creations.
4. The paper would benefit from a clearer claim. The original claim presented in the beginning of the paper (that it shows how aesthetic experience is just as relevant as the creeds in Catholicism) does not seem to be what the rest of the paper actually is about. The rest of the paper presents the importance of aesthetic experience for Christian interpretation, to be sure. However, it seems the thesis is doing something deeper than this original claim.
Author Response
Thank you for your comments.
(1) Why "only" Catholicism? Because the stated aim of the paper is to discuss the incarnational-sacramental imagination that is central to Catholic belief and practice, and to demonstrate that this imagination works beyond text-based artifacts of the tradition. That this imagination can be operative in other Christian traditions is not denied, but I take your point that this could be made clearer. I do so in my revision of the section on Rembrandt and Caravaggio. The use of Merleau-Ponty's phenomenology of perception is crucial to addressing the openness of embodied subjectivity to the "truth of the world" and how that is necessarily consituted (not created) through the agency of both subject and world. It is also crucial for addressing the chiasmus (M-P's term) of the visible and the invisible (here, also the mediation of the infinite by the finite, nature by grace).
(2) To respond to this would take another full paper in order to lay out an aesthetic theory and to argue from this for the inclusion of music and other performing arts (other ways of transcending the linguistic paradigm that bedevils theological reflection). This particular essay, in an issue on religion and aesthetics, makes assumptions that the audience will notice the emphasis on the visual arts.
(3) My use of both Mitchell and Steiner at the outset replies to some of this. Of course, images have different effects. But, as recent research shows (and as Benjamin already predicted), the image-storm of the past four decades or so and the power of late capitalism/neo-liberalism have flattened out the power of images to make them commodities. Mitchell and Steiner (and others like Belting, Freedberg, and Nehamas) attempt to delve deeper to discover how and why images still have power that outruns commoditization.
(4) I have clarified my thesis in the revision.
Reviewer 3 Report
I am, as a reviewer, devoted to the arts and to theology. In that sense, I am quite sympathetic with this article’s goals. But in another sense, I do not agree with the thesis of this essay — which is that art is as important as dogma for Catholics — and I am a tough audience for it. For I do not think these are equal or the same: the affirmations of dogma, which are intellectual judgments to which the mind assents in faith, and the material of artistic symbol, which are material-intellect but not affirmations of specific judgments. They are different, and important differently, which is why we need them.
I have to begin first by saying that “Five Looks At Emmaus” does not understand Catholic sacraments (esp. pp. 2-4) in any “thick” sense at all, and does not seem conversant with the sacraments in a Patristic, Medieval, or Modern sense. Sacraments are, Thomas Aquinas insists, visible signs of invisible grace, and, as signs, they are always accompanied by words, or else there is nothing to complete them: the symbols in the signs are too broad, and grace remains unspecified. There is, in other words, more than a performance occurring in the sacraments, and rather more than the presence-ing of grace in an artistic fashion. In fact, early Christians were so keen to distinguish the liturgy from other public events that they deeply resented the performances of theatre and games.
Then there is a theological collapse between the meaning of Nicaea (whose creed is offered as nothing more than a gloss on artistic, incarnational conviction) and the broader goal of making a point about how images / or bodies / or matter / or art (are these the same?) all evoke Christ and Christianity and sacrament. Their connections therefore rely on their ambiguities and emptiness of meaning.
“Five Looks At Emmaus” is at once an essay too much and too little. Its exploration of its art (mostly post-Reformation) is probably the most interesting and strongest in the current version, though it doesn’t have a constrained sense of time or place, artistic influences, painting mediums (the technology of color production, for example), and other historical points of references. The article very much gives the feeling of floating across time at will, leaping as it does from Nicaea to the Baroque, from artist to artist. I do not think this aids its point, since it seems unable to deal with complications in its argument. The various iconoclasms that sit scattered across Christian history do not emerge as any counter-point, let alone do the complexities of artists like Caravaggio, the brawler-murderer with such a sensitive eye for God.
I would wish that some other version of this article focused itself. It should in some way constrain both its thesis and its material explorations, so that a reader can have both a more precise and a less grand view. A smaller thesis that dares something less than identity with dogma (which I will get to) would be, I think, more successful. It could be the first stone set in a much larger path crossing many arguments and issues.
But if Catholic art is in some special way incarnational, then surely something other than its religious feeling matters about it: surely history matters, and colors, and places? And what is so special, apart from the numinous feeling of the transcendent God? Why is Muslim art, with its geometric expansiveness, somehow less an affirmation of this feeling? Is it really in fact less? These are artistic and theological problems the article provokes but cannot answer, because it is busy treating a broad theme as rather more self-evident than it is.
The author of this article should do more theological or historical reading, or focus on some particular feature of art that they would like to explain and connect to the Eucharist.
It needs revision in its language but is mostly clear.
Author Response
This reviewer has missed the major points of the essay: (a) to show that the incarnational-sacramental imagination that structures Catholic belief, reflection, and practice has a wider definition and application than the seven dogmatically-defined sacraments (and is indeed their presupposition); (b) that the faith-claim that nature mediates grace/the finite mediates the infinite exceeds the text-based artifacts of the tradition and is supported by experiences and traditions that do not fit the linguistic paradigm, including the aesthetic.
This demonstrates how the ensemble of practices, reflections, and beliefs that make up the Catholic (Christian) tradition rely on a "history of effects" (Gadamer's Wirkungsgeschichte) that includes more than a propositional model of theological reflection.